


# 300-years of hydrological records and societal responses to droughts and floods on the Pacific coast of Central America

Alvaro Guevara-Murua [1,2], Caroline A. Williams[2,3], Erica J. Hendy [1,2], Pablo Imbach[4]

[1] School of Earth Sciences, Queens Road, University of Bristol, Bristol BS8 1RJ, UK

[2] Cabot Institute, Royal Fort House, University of Bristol, Bristol, BS8 1UJ, UK

[3] Department of Hispanic, Portuguese and Latin American Studies, School of Modern Languages, University of Bristol, Bristol BS8 1TE, United Kingdom

[4] Climate Change, Agriculture and Food Security Program (CCAFS). International Centre for Tropical Agriculture (CIAT) Hanoi, Vietnam.

Correspondence to: A. Guevara-Murua (alvaro.guevara.2012@my.bristol.ac.uk)

**Abstract.** The management of hydrological extremes and impacts on society is inadequately understood because of the combination of short-term hydrological records, an equally short-term assessment of societal responses and the complex multi-directional relationships between the two over longer timescales. Rainfall seasonality and interannual variability on the

Pacific coast of Central America is high due to the passage of the Inter Tropical Convergence Zone (ITCZ) and large-scale phenomena El Niño Southern Oscillation (ENSO). Here we reconstruct hydrological variability and the associated impacts drawing on documentary sources from the cities of Santiago de Guatemala (now Antigua Guatemala) and Guatemala de la Asunción (now Guatemala City) over the period from 1640 to 1945. Near continuous records of city and municipal council meetings provide a rich source of information dating back to the beginning of Spanish colonisation in the 16thC. Beginning

in 1640, we use almost continuous sources, including >190 volumes of *Actas de Cabildo* and *Actas Municipales* (minutes of meetings of the city and municipal councils) held by the *Archivo Histórico de la Municipalidad de Antigua Guatemala* (AHMAG) and the *Archivo General de Centro América* (AGCA) in Guatemala City. For this 305-year period (with the exception of a total of 11 years where the books were either missing or damaged), information relating to Catholic rogation ceremonies and reports of flooding events and crop shortages, were used to classify the annual rainy season (May to

October) on a 5 point scale from very wet to very dry. In total 12 years of very wet conditions, 25 years of wetter than usual conditions, 34 years of drier conditions and 21 years of very dry conditions were identified. An extended drier period from the 1640s to the 1740s was identified as well as two shorter periods (the 1820s and the 1840s) dominated by dry conditions. Wetter conditions dominated the 1760s-1810s, possibly coincident with reconstructions of more persistent La Niña conditions that are typically associated with higher precipitation over the Pacific Coast of Central America. The 1640s-1740s

dry period coincides with the onset of the Little Ice Age and the associated southward displacement of the ITCZ.



## 1. Introduction

Inter-annual precipitation in Central America is highly variable and has extreme socio-economic consequences (Magrin et al., 2014). The region is considered one of the most sensitive tropical regions to climate change, particularly through hydrological cycle impacts (Magrin et al., 2014; Giorgi 2006). Rainfall anomalies are expected to impact stream flow and
freshwater supplies, agriculture and food production (Hidalgo et al., 2013; Magrin et al., 2014). Additionally, extreme weather events lead to natural hazards, such as floods, landslides and droughts, with a cascade of social consequences (Magrin et al., 2014). For example, the strong drought of 2014-16 has left more than 2 million people in Central America struggling to feed themselves due to poor harvests (WFP, 2015; 2016). At the other extreme, the flooding and winds caused by Hurricane Mitch in 1998 killed more than 11,000 people in this region and caused damage estimated at more than $5
billion (Lott et al., 1999). Projections of increased drought in Central America in the next few decades (Magrin et al., 2014) are, however, less certain as a consequence of the short record of 20$^{th}$ century instrumental observations, high rainfall variability, and the lack of any significant trend in total precipitation over the last 50 years across the region (Aguilar et al., 2005; Marengo et al., 2014). It is therefore necessary to extend the climate record in order to better understand the mechanisms that govern the hydrological cycle and extreme weather events.

From the 16th to the early 19th century the Kingdom of Guatemala formed part of Spain's colonial empire and was governed at regional and local level by a hierarchy of Spanish officials, of whom the most senior was the President of the *Audiencia*, resident in Guatemala but with oversight over a vast area encompassing present-day Belize, Guatemala, El Salvador, Honduras, Nicaragua and Costa Rica (MacLeod, 2010). The system of colonial administration established by the Spanish
Crown during this period generated an extensive documentary record of exceptional value for reconstructing rainfall variability, tropical cyclones and other extreme weather events and their impacts, including on agriculture and the food supply, and on the health of the region's populations. The first reconstruction of rainfall and extreme weather events in Guatemala from documentary sources was by Claxton (1986), revised in 1998. Both studies qualitatively described flood and drought periods from the 16$^{th}$ to the 20$^{th}$ century based on secondary sources such as Lutz (1976) and Pardo (1944), and
discontinuous primary sources consisting mainly of records of tax exemptions (*exoneraciones de tributos*), supplies (*documentos de abastos*), local newspapers and magazines, official reports of the Ministry of Public Works and Agriculture and in some instances information derived from personal correspondence with scholars of Guatemala. Claxton obtained additional information for some years from city council records.

The significance of colonial records relating to the activities of city councils (*Actas de Cabildo*) for complementing modern climate change biased instrumental records is now recognised by climate scientists (e.g., Prieto and García-Herrera, 2009) and reconstructions of rainfall and extreme weather events have been undertaken for regions of Latin American and Spain using both these and ecclesiastical records such as *Actas Eclesiásticas* (e.g., Garza Merodio, 2002; Terneus and Gioda, 2006,





Domínguez-Castro et al., 2008, Cuadrat Prats, 2012). The former are of particular importance as councils dealt with local concerns on a day-to-day basis, and thus provide a continuous and consistent record of climatic information that is detailed, reliable, and can be quantified and dated precisely (e.g., Martín-Vide and Vallvé, 1995; Domínguez-Castro et al., 2008). *Actas de Cabildo* of the capital of the Kingdom, Santiago de Guatemala (Antigua Guatemala) are available until 1775, and

for Nueva Guatemala de la Asunción (Guatemala City) for the years thereafter. For the period following independence from Spain in the 1820s, this source can be supplemented with the records of municipal councils (*Actas Municipales*) of Guatemala City until 1860, and from Antigua Guatemala until 1945, allowing us to extend the reconstruction of rainfall variability into the 20$^{th}$ century.

Both capitals of the Kingdom of Guatemala, Antigua Guatemala and Guatemala City, are in the Dry Corridor (DC) of Central America, an ecological area defined by the Central American dry tropical forest that starts in Chiapas (southern Mexico), and extends across the central and lowland areas of the Pacific coast of Guatemala, El Salvador, Honduras, Nicaragua, to the Pacific province of Guanacaste in Costa Rica (Fig. 1; Arias et al., 2012). This region, which holds most of the Central American population (Imbach et al., 2015), not only shares a common vegetation, but also a similar climate and

rainfall variability (Fig. 1), presenting lower precipitation than many other areas of Central America, and more severe and prolonged dry periods (Arias et al., 2012). Thus, the common precipitation regime along the Pacific coast of Central America (Fig. 1) makes the climatic proxies held in the *Actas* of Antigua Guatemala and Guatemala City especially useful for understanding the hydrological cycle of the wider region and the mechanisms that have controlled it during the last 300 years. Here we construct a new semi-quantitative hydrological index valid for the Pacific coast Dry Corridor using the *Actas*

*de Cabildo* and *Actas Municipales* of Santiago de Guatemala and Guatemala de la Asunción over the period 1640-1945. We also explore the various processes controlling inter-annual and inter-decadal hydrological variability over this region using our extended reconstruction.

## 2 Study area

**2.1 Climatological characteristics of the Pacific coast of Central America**

Precipitation on the Pacific coast of Central America is characterised by a well-defined rainy season following the annual migration of the Inter Tropical Convergence Zone (ITCZ, Portig, 1965; Hastenrath, 1967). In Guatemala, this rainy season is from May to October (Fig. 2 and 3), ending when the ITCZ migrates south (Portig, 1965; Hastenrath, 1967). During the Northern Hemisphere summer, easterly waves and tropical cyclones travel across Central America with the northeast trade

winds, bringing precipitation to the region (Hastenrath, 1967). Storms, or *temporales* in Spanish, are disturbances that originate in the Pacific ITCZ, especially between September-October (Hastenrath, 1976), and are responsible for a substantial part of the annual precipitation of the Pacific Coast of this region (Hastenrath, 2002). In common with the



western side of Central America, the rainy season in Guatemala has maxima in precipitation in both June and September, and a minimum in July and August, named as the Mid Summer Drought (MSD; Fig. 2 and 3; Magaña et al. 1999).

Hastenrath (1967; 2002) proposed that one of the main mechanisms causing the MSD is the northward movement in the early summer and the southward movement in the late summer of the Pacific ITCZ. However, the fact that the time frame of the MSD remains constant at different latitudes, and that the MSD occurs as far as 20°N, where the ITCZ does not cross twice during the rainy season (Waliser and Gautier, 1993) supports Pacific sea surface temperatures (SSTs) and trade winds fluctuations as the more likely cause of the MSD (Magaña et al. 1999). The start of the rainy season (May-June) occurs when the eastern Pacific warm pool (area that corresponds to the average 28°C isotherm from 100°W to the west coast of Central America and Mexico, and from 18°N to 5°N; Karnauskas and Busalacchi, 2009) SSTs exceed 29°C. Later, in July-August, SSTs over this region decrease by ~1°C due to diminished downwelling insolation associated with greater cloud cover in the early part of the rainy season, and consequently leading to a decrease in both convection and precipitation over land (Magaña et al., 1999). Coincident with the decrease in convective activity over the Pacific, the cyclonic circulation present during the first part of the rainy season weakens (Magaña et al., 1999). This atmospheric pattern strengthens the trade winds over Central America, resulting in strong convection and precipitation over the Caribbean side of this region, and subsidence and dry conditions over the Pacific Coast (Magaña et al., 1999). The opposite phenomenon occurs during the late part of the rainy season, when SSTs in the eastern Pacific warm pool increase to 28.5°C because of the stronger solar radiation during the MSD and weakened trade winds (Magaña et al., 1999). This is translated into a strengthening of low-level convergence and deep convection, which generates a second maximum in precipitation in early autumn (Magaña et al., 1999).

On an inter-annual time scale, the Tropical North Atlantic SSTs (Aguilar et al., 2005) and ENSO (Fig. 2, Magaña et al., 2003) play significant roles in the precipitation of the Pacific coast of Central America. Positive North Atlantic SSTs enhance local evaporation and consequently increase water vapor transport to Central America by the easterly trade winds (Aguilar et al., 2005; Hastenrath and Polzin, 2013; Polzin et al., 2015). SSTs in the tropical Atlantic can also increase precipitation over Central America by strengthening the Atlantic TCs season (Aguilar et al., 2005). On the other hand, positive SSTs in the eastern tropical Pacific, associated with El Niño, decrease rainfall over the Pacific coast of Central America (Fig. 2, Magaña et al., 2003; Hastenrath and Polzin, 2013; Polzin et al., 2015). Warm ENSO conditions lowers the number of tropical cyclones over the North Atlantic Basin due to an increase in vertical wind shear and weaker easterly waves (Gray, 1984; Goldenberg and Shapiro, 1996; Goldenberg et al., 2001). El Niño also strengthens the easterly Caribbean low-level jet due to a negative SST gradient from the Pacific to the Atlantic, which results in an enhanced pressure gradient over Central America (Enfield and Alfaro, 1999; Rauscher et al., 2008; 2011) and more subsidence along the Pacific coast (Magaña et al., 2003). In addition, El Niño years are related with a southward position of the ITCZ (Waliser and Gautier, 1993) due to a weaker SST meridional gradient between the eastern Pacific cold tongue region and the Mexican coast warm pool, bringing drier conditions to this region (Magaña et al., 2003). The strongest reduction in precipitation during El Niño



occurs during the MSD prior to the peak of a warm ENSO event, which usually occurs during boreal late autumn and winter (NDJ; Fig. 2; Trenberth, 1997; Curtis, 2002). Inversely, La Niña increases rainfall over the Pacific coast of Central America due to the opposite factors mentioned above (Fig. 2; Curtis, 2002; Magaña et al., 2003).

**2.2 History, setting and climate of Antigua Guatemala and Guatemala City**

The cities of Santiago de Guatemala (Antigua Guatemala) and Nueva Guatemala de la Asunción (Guatemala City) are located 35 km apart in the south-central region of the Republic of Guatemala, at an altitude of approximately 1500m. The cities share a similar temperate climate and have an annual average precipitation of approximately 1100mm for Antigua Guatemala and 1200mm for Guatemala City, with monthly distribution following the bimodal pattern of the Pacific coast of

Central America (Fig. 3). The Pensativo Basin (Fig. 1), circumscribed on the east side by the Guacalate basin and part of the bigger Achiguate river catchment, where Antigua Guatemala is located, is characterized by steep slopes and sediments with high permeability (Quiroa-Rojas, 2004) and consequently the river flow is intermittent and typically ceases during the dry season (Fig. 3). High discharge rates and sediment load in response to the rainy season have caused regular problems for the population of the area of Antigua Guatemala, including severe flooding (Quiroa-Rojas, 2004).

These two cities, successive capitals of the Kingdom of Guatemala, contained a significant proportion of the Spanish population in Central America during the period of study (MacLeod, 2010). Santiago de Guatemala (Antigua Guatemala; MacLeod, 2010) was founded in 1543, after a strong mudflow from Agua volcano in September 1541 destroyed the previous capital, today known as Ciudad Vieja (Hutchison et al., 2014). Santiago became the most important city in Central America

(Lutz, 1997) and after 1773, following an earthquake of an inferred magnitude of 7.5 (Villagran et al., 1996) that destroyed most of the city, the capital was relocated again, this time to the Valley of the Ermita, and named Nueva Guatemala de la Asunción (today Guatemala City).

**3.  Data sources and Methodology**

Minutes of the meetings of the city and municipal councils (*Actas de Cabildo* and *Actas Municipales*) of Santiago de

Guatemala (Antigua Guatemala) and Guatemala de la Asunción (Guatemala City) provide information on the responses of the communities and the city and municipal councils to hydrological stresses, among other issues of concern to the local community. In the following section we describe these direct and indirect indicators, provide examples from the documents, and outline how we have used these to reconstruct a continuous semi-quantitative indicator of hydrological stress for both cities and the surrounding area from 1640 to 1945.






### 3.1 Indicators of hydrological variability

Direct information in the *Actas de Cabildo* and *Actas Municipales* consist of descriptions of significant meteorological events: heavy rains, flooding events, dry conditions and droughts. As heavy rain and flooding are sudden events, their descriptions in the records usually occur within a period of no more than a few weeks of the event. Information on droughts

and dry periods extends over longer periods, and are often linked to descriptions of crop loss or shortages and how these were dealt with at the local level. Indirect indicators are primarily descriptions of the effects of extreme weather events on the population, such as famines, food shortages, or structural damage caused by excessive rainfall or flooding. We only use indirect information as a proxy to reconstruct rainfall variability where the mechanism that creates the problem for the population (e.g., a flooding event that destroys a bridge; lack of rain that causes food scarcity) is clearly stated in the *Actas* or

in other historical sources. This conservative approach is taken because excessive rainfall and drought can have the same result – a deficit of crops, despite opposing causes. In addition, there are a few accounts within the city and municipal council records which mention either hunger or grain scarcity attributed to artificially inflated prices and caused by speculation and hoarding by grain merchants. Thus, indirect information needs to be assessed carefully to establish whether a climatic event is responsible.

A key additional source of climatic information during the colonial period is the rogation ceremonies carried out by municipal and church authorities. A rogation ceremony, known as a *rogativa* in Spanish, was a Catholic religious ceremony of prayer or supplication which frequently took place during a period of anomalous conditions, such as when a meteorological event affected the agricultural cycle or in the aftermath of a natural disaster (Martín-Vide and Vallvé, 1995). The central role of the Catholic Church in Spanish culture meant that a religious act was considered an effective means to

prevent or relieve the effects of natural disasters (Martín-Vide and Vallvé, 1995) and was common practice in Spain (e.g., Martín-Vide and Vallvé, 1995; Rodrigo and Barriendos, 2008; Domínguez-Castro et al., 2008; Cuadrat Prats, 2012), across the territories that once formed part of the Spanish Empire (e.g., Garza Merodio, 2002; Terneus and Gioda, 2006), and in other Catholic countries such as Italy (e.g., Piervitali and Colacino, 2001). Rogation ceremonies are also considered a useful climatic proxy to reconstruct rainfall variability because they are relatively consistent over time due to strict ecclesiastical

rules and because they were always recorded in documents such as the *Actas* due to the costs involved in holding the ceremonies (e.g., Martín-Vide and Vallvé, 1995; Piervitali and Colacino, 2001; Rodrigo and Barriendos, 2008; Cuadrat Prats, 2012).

Rogation records as indicators of hydrological stress in Guatemala need cautious analysis. After the first quarter of the 18[th]

century, as noted in the *Actas*, the authorities began to conduct annual rogation ceremonies to the *Virgen Nuestra Señora del Socorro* (Our Lady of Socorro). These ceremonies (or *novenarios*) to pray for good rains took place over 9 days at the beginning of May (AGCA, 1723, sig. A1.2.2, leg. 1790, Exp. 11784, folio 26). The timing of the *novenario* may be related to the fact that May is the month of Mary in the Catholic calendar, but the entries in the *Actas* suggest that in Guatemala its



significance owed more to the fact that May coincides with the start of the critical rainy season. Because the *novenario* was undertaken as preventative insurance rather than in response to immediate weather conditions it cannot be used as an indicator of dry conditions. Rogation ceremonies made prior to the 1720s or in addition to the *novenario*, however, are clear indicators of a lack of rain and we use records of these specific ceremonies to reconstruct hydrological drought conditions.

The Claxton (1986; 1998) compilation of drought and flood episodes in Guatemala starting from the middle of the 16[th] century, based on discontinuous primary documentary and secondary sources, is used in this study to corroborate the information obtained from city and municipal council records and to fill any gaps associated with the way in which the information was recorded in the *Actas*. In conjunction with Claxton (1986; 1998), we also use Pardo's (1944) compilation of

notable events in Antigua Guatemala to help us to categorize years in which city or municipal council records indicate food shortages but do not attribute these to either an over abundance or lack of rainfall. However, we have excluded extreme weather events reconstructed by Claxton (1986; 1998) where these are based on personal correspondence from our reconstruction as we are unable to cross-check with other sources.

**3.3 Example entries from city and municipal council records**

The climatic information contained in the *Actas* of Antigua Guatemala and Guatemala City closely follows the hydrological cycle and the characteristics of the rainfall season of this region. The *novenario*, conducted annually in May from the first quarter of the 18[th] century, can be considered to mark the start of the rainy season, as the following entry for the year 1817 shows:

… we approach the time [of year] when this Illustrious City Council customarily celebrates a *novenario* of masses to Our
        Lady of Socorro, carries her image in procession [through the streets] and implores through her intercession that there be
        no shortage of rain during the [rainy] season … (AGCA, 1817, sig. A1.2, leg. 2192, Exp. 15743, folio 40).

If the rainfall season was wetter than usual, the *Actas* usually contain information about small-scale surface flows, abundance of rain, etc., as in this 1727 response to an enquiry regarding:

the measures that may be taken to repair the bridge of Los Esclavos, which appears to have been ruined as a result of
        the winter (weather) and the copious amounts of water that have flowed through [the bridge] … (AGCA, 14/11/1727,
        sig. A1.2.2, leg. 1791, Exp. 11785, folio 49).

In the most extreme cases, such as severe flooding, the *Actas* contain a description of the specific event, the damage caused,

and a record of the responses of both the council and the city's inhabitants. Most flooding events occurred during the late rainy season, and especially in September and October, in accordance with a higher Pensativo average river flow (Fig.3), such, for instance, was the case in 1697:




> This council [meeting] discussed the news … from the towns of San Juan Gascón and San Miguel el Alto … to the effect that the winter season having been so harsh this year, and the floods severe, a great portion of the main [water] tank collapsed at the point where two streams converge and flow [into it]; it was agreed that the tank should be repaired … (AGCA, 08/10/1697, sig. A1.2.2, leg. 1785, Exp. 11779, folio 117).

However, significant flooding events also happened in other periods of the rainy season, such as in 1878:

> [Heavy] rain has caused serious damage along the Pensativo River. The riverbed having silted up it flowed over properties situated along its banks. Establishing the means to prevent such damaging events [in the future] is a matter of the utmost importance; accordingly … an open meeting of the municipality will be called to identify the most appropriate measures to address this ongoing threat …(AHMAG 10/07/1878, Actas de las Sesiones Municipales, folio 58).

During some years, the *Actas* also contain information on the impact of excessive precipitation during the rainy season:

> The copious rains have been the reason why, in various districts, maize harvests have been scarce … and why it is feared, with good reason, that  this capital will suffer shortages of said grain. This calamity should have and could have been anticipated, had regional officials been vigilant in monitoring the situation and made provision for bringing more land
under cultivation which is their primary obligation; but they have not complied, nor have they even informed this government of the state of the harvest in their respective regions. … (AGCA, 02/1817, sig. A1.2.2, leg. 2192, Exp. 15743, folio 55).

Prior to the 1720s, if rainfall during the first part of the rainy season was irregular and lower than usual, the city celebrated a
rogation ceremony to the Virgin of Our Lady of Socorro in May. On occasion these records also include information about the state of fields and crops:

> [The meeting] discussed the fact that although the season for rain is well advanced, … we have experienced such a lack of it that as of today it has not rained [at all], resulting in heat so excessive that it is feared that it will have harmful consequences, and that lack of water will also lead to poor harvests, assuming there are any. It was [in consequence]
resolved … that a [service of] rogation would be held before the most holy image of Our Lady of Socorro, in her chapel in the Holy Cathedral Church of this city on Saturday 25th of this month (AGCA, 23/05/1709, sig. A1.2.2, leg. 1787, Exp. 11781, folio 100).

If the situation was particularly severe, a rogation ceremony to Our Lady of Socorro was carried out in the middle of the
rainy season (July, August or September), as the following example from 1699 shows:

> In this council [meeting] it was agreed that due to the lack of rain that … is causing the severe illnesses which are afflicting this city, … and which also threatens the failure of the critical wheat and maize harvests … the city magistrates will request … licence to hold a procession of rogation to the most holy image of Our Lady of Socorro, [the procession to end] in the church of the convent of our Lady of Mercy. To this end, they are to … make … arrangements … for the





procession … as well as for a sung Mass to be celebrated on the day following … (AGCA, 28/07/1699, sig. A1.2.2, leg. 1785, Exp. 11779, folio 206).

As rogation ceremonies incurred substantial costs, they were only conducted when drought was severe and its impact significant (Martin-Vide and Vallvé, 1995). These ceremonies are much more common in the record before the tradition of the *novenario* began in the first quarter of the 18th century, and appear to have ceased entirely by the 1820s. From the 1820s onward evidence of drought conditions is identified from discussion of the state of the harvests or the availability of basic foodstuffs. In 1914, for example, the desirability of a committee being established was discussed, specifically to address 'the difficult circumstances the country is currently facing due to the lack of rain', and the necessity of acquiring basis foodstuffs,

implying significant shortages (AHMAG 28/08/1914, Actas de las sesiones municipales, folio 67-68).

On occasion, information about crop shortages or harvest failures or flooding events is recorded in the first few months (dry season) of the following year's book of *Actas*, which allows the reconstruction of hydrological conditions for the previous year.

### 3.4 Definition of the semi-quantitative hydrological index

A five-point scale (from -2 for very dry to +2 for very wet; Table 1) was developed to describe the hydrological characteristics of each year based on information from the wet season (May-October; accounting for >90% of the annual rainfall), which follows the approach used previously to reconstruct precipitation variability from documentary sources (e.g., Nash and Endfield, 2002; Prieto, 2007; Berland et al., 2013). The 305-year reconstruction draws on information extracted from a total of more than 190 manuscript books held by the *Archivo General de Centro America* (AGCA, Guatemala City)

and the *Archivo Histórico de la Municipalidad de Antigua Guatemala* (AHMAG, Antigua Guatemala). This information was converted into annual indices, based on precipitation intensity, severity of shortages, flooding intensity, rogation records, and the time of year in which the reported event or circumstances occurred. A year is classified as normal (0) if no rogation ceremonies were recorded in the *Actas* after May (or during May before the custom of the annual *novenario* commenced in the first quarter of the 18th century), and no indirect or direct climatic information was provided in the records (absence of

information is interpreted to mean absence of anomalous climatic conditions).

Accounts of droughts and inadequate rains from Claxton (1986; 1998) and Pardo (1944) were also used to reconstruct 5 dry years (between 1640-1840) where crop shortages are mentioned in the *Actas* but without any indication as to their cause. In addition, due to the fact that less information about drought conditions and crop shortages is recorded in the *Actas* of Antigua

Guatemala and Guatemala City through time (after 1840 only the strong drought of 1914 and crop shortages in 1860 are explicitly recorded in the *Actas*), 11 of a total of 13 years for the period 1840-1945 characterized by dry conditions are based on Claxton's (1986; 1998) information only. All these years are characterized as -1.





### 3.5 Instrumental observations

The instrumental records from three meteorological stations located in the vicinity of Antigua Guatemala and Guatemala
City, and held by the Instituto Nacional de Sismología, Vulcanología, Meteorología e Hidrología (the National Institute of
Seismology, Volcanology, Meteorology and Hydrology), are compared with the overlapping period in the rainfall
reconstruction (1910-1945).  Two sets of rainfall records are from the area of Antigua (El Potrero, 1910-71; and Antigua,
1934-71) and the third is located in Guatemala City (INSIVUMEH, 1928-12). An additional 6 years from 1879-1882 and
1896-1897 reported in Guatemala City are also compared with respect to the reconstructed hydrological index. Estimated
monthly runoff data per month, for the period 1981-2010, derived from the land surface model Orchidee and based on
precipitation observations is included in Fig. 3 to show how the flow of Pensativo river varies during each month.

## 4. Results

### 4.1 Hydrological reconstruction for Guatemala City and Antigua Guatemala

The books of *Actas de Cabildo* and *Actas Municipales* were analyzed chronologically to produce an almost continuous
record of hydrological variability affecting Antigua Guatemala and Guatemala City for the period 1640-1945 (Fig. 4). Only
11 years (1776, 1782, 1785-1786, 1797-1798, 1830, 1855, 1858, and 1861-1862) could not be reconstructed because the
books were missing or damaged. Our analysis revealed 12 years of very wet conditions, 25 years of wet conditions, 34 years
of dry conditions, and 21 years of very dry conditions.  Dry conditions persisted in the 1640s-1740s, 1820s, and 1840s (Fig.
5), including some exceptionally dry decades when 4-5 years were recorded as dry (1660s, 1720s, 1730s and 1820s; Fig. 5).
The 1760s-1810s seem to be dominated by wetter conditions, characterised by a low number of dry years and a relatively
high number of wet years, including some major flooding events or crop shortages due to abundance of rain, such as in 1762,
1783 and 1792. The 1780s and 1810s were the decades with the highest number of wet years (3 and 4 wet years per decade
respectively), whereas the 1690s and 1890s are notable for their high variability between extremes of both drought and
25   flooding (Fig. 5).

### 4.2 Reconstruction verification

In the overlapping 35-year period (1910-1945; Fig. 6) the driest (1914) and wettest (1933) years of the 20[th] century recorded
by three nearby meteorological stations are correctly classified in the documentary-based hydrological reconstruction (Fig.
6). A further two dry years (1912 and 1920) and two wet years (1921 and 1944) also match the anomaly recorded from
30   instrumental observations, but two years (1911 and 1923) categorized as wetter than average in the index were drier than



average according to the El Potrero rainfall observations. In addition, the wetter conditions of 1896-1897 according to instrumental observations from Guatemala City are picked up in the *Actas,* including the wettest month of the entire 7 years dataset (June 1897; 422.5mm), when the minutes of the city council mention flooding caused by the Pensativo river (AHMAG, Libro de Actas n. 65, Folio 98-99). On the other hand, the years of 1879-1882, which were also wetter than

normal according to instrumental observations, are classified as normal in the hydrological index, based on absence of hydrological information.

**5. Discussion and Conclusions**

A near-continuous reconstruction of hydrological variability from 1645 onwards for the Pacific coast of Central America
was derived from documentary sources collected from the *Archivo General de Centro América* (AGCA, Guatemala City) and the *Archivo Histórico de la Municipalidad de Antigua Guatemala* (AHMAG Antigua Guatemala). In addition the *Actas* are a valuable source of information for understanding how the society of the Pacific coast of Central America was affected by extremes in hydrological conditions and their response to inter-annual to inter-decadal variability over a 300-year period. We highlight this information first, before demonstrating the value of these historical sources for climate reconstruction. We
conclude with an analysis of the decadal-century scale variability identified including the severe and persistent dry conditions in the 1640s-1740s, 1820s, and 1840s  (Fig. 4 and 5), and contrast with the 1760s-1810s which include the wettest decades of the reconstruction (1780s, 1810s; 3 and 4 wet years per decade respectively) and a low number of dry years relative to the rest of the reconstruction.

**5.1 Societal impacts and responses to past extreme-weather events**

During the 17[th] and 18[th] century, when dry years were much more common than later centuries (Fig. 5), these conditions were linked in the *Actas* to increased rates of illness, partly as result of food and water scarcities and partly due to the increased heat (the latter a result of reduced cloud cover and increased shortwave radiation). In response the city council arranged rogation ceremonies to pray for good rains which in turn would bring an end to illnesses affecting the population,
as stated in the city council records of 1701, for instance:

This [meeting] of the council discussed the fact that due to the lack of rain [the city] is experiencing an intense heat harmful to public health, such that a widespread epidemic had to be feared, as lack of pasture was leading to the death of livestock, and it was also reported that the crops had not borne fruit. [Thus] … it was resolved … that the city would appeal to the Holy Virgin Mother of God … carrying out a rogation ceremony to and procession of the
miraculous image of Our Lady of Socorro. (AGCA, 19/05/1701, sig. A.1.2.2, leg. 1786, Exp. 11780, folio 22-23).





As further mitigation measures, the city council ordered second sowings to be made in other regions with a more favorable climate, such as Escuintla, nearer to the Pacific coast, where both rainfall (average precipitation for the city of Escuintla is 2800mm) and temperatures (annual average temperature of 26°C) are higher and more conducive to harvest development. The year 1840 is a clear example of how important second sowings were for addressing food scarcity caused by the lack of
rains and high evapotranspiration:

> The *corregidor* [district official] has sent another note reporting that due to the lack of rain, most of the lands sown with grains for the supply of this city are in a poor state and almost ruined, that this [situation] requires prompt and effective action, and that he hopes that the municipality, taking into consideration the critical importance of the matter, will issue instructions … for the inspection of fields and granaries in the vicinity [of the city], and in the event [that these are found
wanting] for additional lands to be brought under cultivation, either in the town of San Pedro Mártir or elsewhere ... (AGCA, 11/08/1840, sig. B78.1, leg. 593, Exp. 10120,NF).

In contrast, wet weather events were commonly associated with flooding of the city of Antigua Guatemala by the Pensativo River. It was recognized that Pensativo floods were not only caused by intense rainfall, but also anthropogenic activities in
the river catchment, specifically the clearing and planting of crops on surrounding hillslopes, leading to soil erosion and silting up of the river bed. In response, the city council of Antigua ordered preventative annual dredging of the Pensativo River prior to the rainy season, as illustrated by an entry from 1877 (frequently repeated) calling on the authorities to arrange for "this important work" to be undertaken, and to make owners responsible for dredging stretches of the river bordering their lands (AHMAG 13/04/1877, Actas de las sesiones municipales, folio 65).

In addition, the municipal authorities of Antigua Guatemala prohibited the clearing and planting of hillslopes near to the Pensativo River. However, the *Actas* indicate that such prohibitions were often breached, it being reported in 1921 that the hillslopes were "worked at night". On this occasion, the council emphasized that "energetic measures" would have to be introduced to prevent the cultivation or clearing of the hillslopes in critical areas to avoid a widespread "disaster" in the
event of a major flooding event (AHMAG 15/07/1921, Actas de las sesiones municipales, folio 116).

However, in years when these prohibitions were enforced in Antigua and the surrounding towns, damage to the city associated with intense wet events was reduced, or even avoided, as this example from 1923 shows:

> … the Council agreed that this municipality should inform the Ministry of Government and Justice that notwithstanding
the exceptionally strong storm that struck on 11, 12 and 13 October, which wreaked havoc across the country and beyond, this city suffered no damage from the Pensativo River, the bed of which did not become silted up as it has in times past. This confirms, and as shown by experience, that the prohibition on clearing and planting on the slopes of the hills that surround this city … has been highly beneficial (AHMAG 13/11/1923, Actas de las sesiones municipales, folio 93-94).



The accounts presented above demonstrate that the society of the Pacific coast of Central America was conscious of the societal impacts of extreme hydrological events, and had identified successful mechanisms to remediate the risk. In addition, the *Actas* show how societies in the past, as today, could modify the cascading effects of extreme weather events, for

example amplifying flooding by altering catchments (Van Loon et al., 2016). A more extensive examination of historical sources, such as ecclesiastical records, hospital records, and records of tax exemptions from the AGCA and other Central American archives would enable a deeper analysis of societal preparation and response to past extreme weather events in this region.

**5.2 Verification of the hydrological reconstruction**

Instrumental observations should be used to calibrate any proxy climate reconstruction including document-based hydrological records (e.g., Rodrigo et al., 1999; Rodrigo, 2008), and an overlap of 30 years for calibration and 30 years for verification has been recommended (Rodrigo et al., 1999). However, we were unable to find any instrumental river flow data for the Pensativo during the reconstructed time period and the earliest continuous instrumental precipitation record near to

Antigua Guatemala starts in 1910 (El Potrero). The document-based index is relatively consistent with these meteorological station precipitation records (1910-1945 i.e. 35 year overlap; Fig. 6) despite the fact that the information inferred from the *Actas* relates to feedbacks and components of the hydrological cycle and catchment pre-condition rather than simply rainfall volume.

A discrepancy arises in the years 1911 and 1923, categorized as wet in the index, which according to the El Potrero rainfall observations were drier than average. In 1923, the El Potrero monthly rainfall data shows that precipitation between May and August was less than half the average for the 1910-1970 period, but rainfall accumulations in October were three times (325mm) the average (119mm) due to a *temporal* or storm. Thus, the convective rainfall associated with a *temporal,* a short, high intensity rainfall event, could induce flooding impacts that were then recorded in the *Actas* even though the event

occurred in a very localized area or during a generally drier wet season. Similarly, how well catchment soil erosion and sedimentation of the river channel was controlled (see the previous account from 1923: AHMAG 13/11/1923, Actas de las sesiones municipales, folio 93-94) would vary the consequences of the Pensativo River flooding. Under such circumstances a year could be classified as wet when in fact, a weak meteorological drought occurred. In the 1910-1945 period only one such short, but strong, wet episode within a dry year occurred, indicating that the number of potential years misrepresented

in the index due to this problem is likely to be low.



In the second case, precipitation for all months of the 1911 wet season recorded at El Potrero was below average, with the exception of May 1911 (198mm with respect to an monthly average of 99mm). However, the municipal council minutes highlight recorded problems in the Pensativo River during September:

The Alcalde presiding informed the Corporation that, as a consequence of the heavy rains of yesterday … the Pensativo River overflowed in the region of the bridge of San Ignacio, and that he thinks that the main damage has been to a rocky section of the riverbed in the *finca* [farm or ranch] of Santa Juez de los Tres Herreros, where the water level is higher, and that removing this obstruction will reduce the risk [of flooding] to the city (AHMAG 08/09/1911, Actas de las sesiones municipales N.--, folio 24).

While heavy rains over the late part of the rainy season, in association with a saturated soil from previous rainfall during the wet season, would explain a higher flow in September (see monthly average flow in Fig.3) and this account in the minutes of the meetings of the municipal council, the monthly precipitation from El Potrero in contrast indicates well below average rainfall in July 1911 (46mm compared to the monthly average of 150mm), August 1911 (48mm compared to the monthly

average of 130mm) and September 1911 (91mm compared to the monthly average of 208mm). Either very localised and concentrated convective rainfall occurred within the Pensativo catchment and so not affecting El Potrero, or there was a problem with the instrumental records.

Of course, it is also possible that some of the dry years according to the instrumental record were recording a deficit in

precipitation (a meteorological drought) that was minor enough not to cause the serious societal consequences characteristic of an agricultural drought induced by soil water deficiency and plant water stress. Climatic information was recorded in *Actas* because it had severe consequences for the population or on the infrastructure. It also makes sense that a less prepared society would be more vulnerable to extreme events and consequently record a greater number of such events. Consequently there is potential for a bias in the index towards wetter conditions with improvements in crop production techniques during

the 19th and 20th century and a reduction in the community's dependence on local crop productions (e.g. faster transportation and more goods available through imports). Additionally, the ending of the rogation ceremonies after the 1820s means that we lose the valuable hydrological information that invariably accompanied the recording of these ceremonies. This fact would result in a reduction of dry years reconstructed from the *Actas* and is another possible source for a bias towards wetter conditions in our hydrological index. To counter these potential issues we use the dry periods reported by Claxton (1986;

1998) to identify additional dry years at the middle and end of the 19th century and early 20th century not recorded in the *Actas*.

Since the Pensativo River has been a persistent threat to the city of Antigua Guatemala, measures of overabundance of rainfall are less susceptible to changes in the way information is reported in the documentary records, especially for those

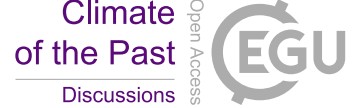



years classified as +2. However, flooding could be exacerbated by the high levels of sediments in the riverbed, making it difficult to distinguish the wet years, classified in this study as +1, from those where problems relating to the dredging of the Pensativo River were at fault and precipitation was not higher than average. Although the most problematic accounts have been excluded from this index, it is still possible that a few records used to categorize a year as wet (+1), could actually reflect anthropogenic caused hydrological problems. Thus, we emphasize that additional documentary sources from this region should be examined to corroborate some of the wet years included in this study. As an example, the account for the year 1932 in the minutes of the municipal council is a clear illustration of a year that, according to instrumental observations, was wetter than average conditions, but which we would not classify as wet on the basis of the information present in the *Actas*, given that the problems associated with the Pensativo as stated in this account could easily be attributed to incorrect dredging of the river:

> The … Alcalde presiding informed the Council that because the riverbed of the Pensativo had silted up, the bridge between Pavón and San Ignacio is now useless, as the water flows over the bridge … as its repair is vital, he requests authority to proceed [noting that] some citizens have offered to contribute [towards costs]. The … Corporation agreed that a budget for the works should be submitted to the next meeting … (AHMAG 20/09/1932, Actas de las sesiones municipales, NF).

Although changes in the way the climatic information was recorded in the *Actas* cannot be excluded as having influenced long-term trends, the significant difference in the number of dry and wet years between the early and late part of the index appears a robust feature.

### 5.3 Inter-annual to centennial-scale hydrological variability

The persistent dry period of the first part of the reconstruction, from the 1640s to the 1740s, contains the highest number of dry years of the reconstruction, peaking in the 1720s-1730s (Fig. 4 and 5). The number of wet years during this period was also lower, with the exception of the 1690s when both the number of dry and wet years was extremely high compared with the rest of the entire period of study. Additionally, of the 32 years between 1640 and the 1820s when we know rogation ceremonies to have been celebrated, 29 occurred during this early period. 16 of the rogation ceremonies took place between the end of April-May, implying an irregular start of the rainy season, while the other 13 were made during the onset of the rainy season (July-September), implying lower precipitation during the early part of the rainy season and stronger drought conditions over the MSD. These specific conditions are characteristic of El Niño-related rainfall anomalies in the Pacific Coast of Central America (Fuentes et al., 2002; Curtis, 2002) and lead to lower annual averaged precipitation (Magaña et al., 2003).



We tested this relationship by comparing our reconstruction with the proxy-based ENSO reconstruction of Gergis and Fowler (2009), and found 35% of dry years coincided with a reconstructed El Niño year, compared to 26% with La Niña. Contrary to expectations, the years in which both the early and late rogation ceremonies were undertaken were no more likely to match a reconstructed El Niño year than a La Niña, and more early rogation ceremonies were celebrated in the years

identified by Gergis and Fowler (2009) as La Niña. This surprising result begs the question whether the recent observations that warm ENSO conditions are related with lower precipitation, an irregular start of the rainy season and a stronger mid summer drought in the Pacific Coast of Central America (e.g., Fuentes et al., 2002; Curtis, 2002; Magaña et al., 2003) is a non-stationary relationship. However, the Gergis and Fowler (2009) ENSO timeseries is also based on a restricted number of proxy records, heavily dependent on the persistence of teleconnections, and so it is important that the analysis of historical

sources is expanded to improve the available reconstructions of past ENSO events.

The persistent dry period from the 1640s to the 1740s was coincident with the period of cooler northern hemisphere (NH) temperatures called the Little Ice Age (LIA, 1500s-1800s; e.g., Bradley and Jones, 1993; Jones et al., 1998; Mann et al., 1998; 1999) and significant glacier advances, including a maximum extent in the South American Andes between the 1630s-1730s (Jomelli et al., 2009). This dry signal has been recognized at other sites in Mesoamerica, including from documentary

sources in Mexico (e.g., Florescano, 1980a; 1980b; Florescano et al., 1980; Swan, 1981; Endfield and O'Hara, 1997). Several authors have suggested an increase in frequency of warm ENSO conditions during the LIA (Cobb et al., 2003; Langton et al., 2008). In addition, strong cooling over the NH, including the North Atlantic basin may have been responsible for the southward movement of the ITCZ during the LIA by an estimated ~ 5° according to proxy evidence derived from lake sediments from the Northern Line Islands, Galapagos, and Palau (Sachs et al., 2009) and lower Ti concentrations in

ocean sediment cores of the Cariaco Basin (Venezuela; Haug et al., 2001). In addition, modeling studies support a southward ITCZ shift when there is a strong cooling episode over the NH, in order to allow more northward heat transport (e.g., Broccoli et al., 2006; Haywood et al., 2013).

The oscillations of the ITCZ (Portig, 1965; Hastenrath, 1967; 2002), and the Pacific (e.g., Fuentes et al., 2002; Magaña et al.,

2003) and North Atlantic SSTs (e.g., Aguilar et al., 2005) are the dominant mechanisms that control precipitation over the Pacific Coast of Central America. Therefore, lower equatorial North Atlantic SSTs (Haug et al., 2001), including the Caribbean region (Winter et al., 2000) and a southward movement of the ITCZ caused by a strong cooling over the NH during the LIA (Haug et al., 2001; Sachs et al., 2009), may have been responsible for the dry conditions that prevailed during the first 100 years of this reconstruction across the Pacific Coast of Central America. This hypothesis is further supported by

the strengthening of the South American summer monsoon (SASM) during the LIA associated with a southward position of the ITCZ found in observation and modelling studies (e.g., Bird et al., 2011; Vuille et al., 2012; Apaéstegui et al., 2014). In addition, the possibility of dominant El Niño conditions during the LIA (Cobb et al., 2003; Langton et al., 2008) may also have played a role, although the question of how ENSO varied through the LIA is still in debate (Yan et al., 2011; Henke et al., 2015; Emile-Geay et al., 2013).



From the 1750s only three decades are marked by a larger number of years with dry conditions, specifically the 1820s, 1840s, and the 1890s, and there is a period (1760-1810s) where wet conditions seem to dominate. This shift to fewer droughts recorded in the *Actas* and wetter conditions on the Pacific Coast of Central America is consistent with the high

frequency of moderate to strong La Niña years between the period 1760s to 1810s (21 events versus 12 moderate to very strong El Niño years during the period 1760-1810s) identified by Gergis and Fowler (2009). However, on an event to event timescale, only 31% (4 of the 13) of the wet years identified in our study for this period occurred during the same year of cold ENSO conditions reported by Gergis and Fowler (2009). Overall, when our entire hydrological index is compared against the La Niña reconstructions from Gergis and Fowler (2009), only a quarter of the wet years are coincident with La

Niña reconstructed years, which is even weaker than the percentage (35%) of dry years coincident El Niño. Interestingly, the percentage of wet events of the early part of the rainy season (May to July) occurring during La Niña year rises to 40%. The Pensativo river modelled flow is lower for the early part of the rainy season than for the later part (Fig. 3), however, high flow rates have been modelled between May to July for the years affected by La Niña, and specifically for 2010. Thus, both the high modelled river flow and the substantial percentage of early flooding events happening during cold ENSO events in

the historical record highlights the possible wetter effects of La Niña on the early rainy season rainfall.

The changes in frequency, extent and severity of extreme weather events (floods and droughts) found in our hydrological index provides valuable information for improving our understanding of Central America's inter-annual and inter-decadal rainfall variability and its sensitivity to climate change. The most plausible explanation for the shift to wetter conditions and

a lower number of dry years is the end of the Little Ice Age, a possible tendency to more La Niña events, warmer equatorial North Atlantic SSTs and a northward movement of the ITCZ. On the other hand, global warming projections suggest that the ITCZ will move southward again by the end of the century, probably related with warm ENSO like conditions within the eastern tropical Pacific (Rauscher et al., 2008; Hidalgo et al., 2013). This scenario could create conditions similar to the 1640s-1740s, because a stronger Caribbean low-level jet and low-level easterlies would cause surface divergence, in

conjunction with an earlier westward amplification and intensification of the North Atlantic Subtropical High would bring drier conditions to the Pacific Coast of Central America (Rauscher et al., 2008). An increase in the frequency of widespread droughts and an earlier start and intensification of the MSD of up to 25% is forecast in some areas of Guatemala, Honduras and El Salvador (Neelin et al., 2006; Rauscher et al., 2008; 2011; Hidalgo et al., 2013). If drier conditions in the future mimic those experienced during the 17th and 18th century, as recorded in city and municipal council records, it would not be

surprising to see an increase in agricultural problems, food shortages and malnutrition for the most vulnerable part of the population. In this climactic scenario Central American's population would not only suffer again the consequences of widespread droughts, but this time, they could be accentuated by the anthropogenic changes made to the land surface over this region, affecting the evapotranspiration, infiltration, surface runoff, water storage and development of drought conditions (Van Loon et al., 2016). Therefore, more than 300 years after the 1640s-1740s dry period, the scenario projected



by general climate models as a consequence of global warming could have stronger socio-economic consequences for the population of this region than in the past, increasing previous and current Central American high vulnerability to climate change.

**Acknowledgements**

This is a contribution to the PAGES 2k Network [through the ###2k working group]. Past Global Changes (PAGES) is supported by the US and Swiss National Science Foundations. This work was supported by a joint U. Bristol postgraduate scholarship and La Caixa Scholarship to A.G.M, and by a RCUK Academic Fellowship to E.J.H. This work was implemented as part of the CGIAR Research Program on Climate Change, Agriculture and Food Security (CCAFS), which

is carried out with support from CGIAR Fund Donors and through bilateral funding agreements. For details please visit https://ccafs.cgiar. org/donors. The views expressed in this document cannot be taken to reflect the official opinions of these organizations. We thank A. Rust, and K. Cashman for their constructive comments to improve the manuscript. The authors also acknowledge support from the Cabot Institute, University of Bristol; Mariano Barriendos Vallvé of the Universitat de Barcelona, Gustavo Chigna of the INSIVUMEH and from Ana Arriola of the *Archivo General de Centro América*.

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

30





**Table 1** Classification of hydrological conditions for the cities of Guatemala and Antigua Guatemala

| Index Value | Rainy season | Rationale |
|---|---|---|
| 2 | Very Wet | Flooding events with severe impacts reported; years during which the city or municipal council records state that crop production was lost due to the abundance of rain |
| 1 | Wet | Records that specifically mention abundance of rain or small flooding events with no severe consequences |
| 0 | Normal Conditions | Records that include any kind of information that the rainy season has developed normally. If no information regarding wet or dry conditions is provided in city and municipal council records, the year is also classified as normal |
| -1 | Dry | Reports of poor harvests due to dry conditions associated with lack of rains according to Claxton (1986; 1998) or Pardo (1944), but not mentioned in city or municipal council records. Rogation ceremonies celebrated in response to the low amount of rain in May (careful consideration is given after the novenario started). Dry conditions reported by Claxton (1986; 1998) after the 1840s. |
| -2 | Very Dry | Records that mention poor crop production multiple times in the book, extending into the following year and caused by dry conditions. Rogation ceremonies carried out due to the low amount of rain in July, August or September |



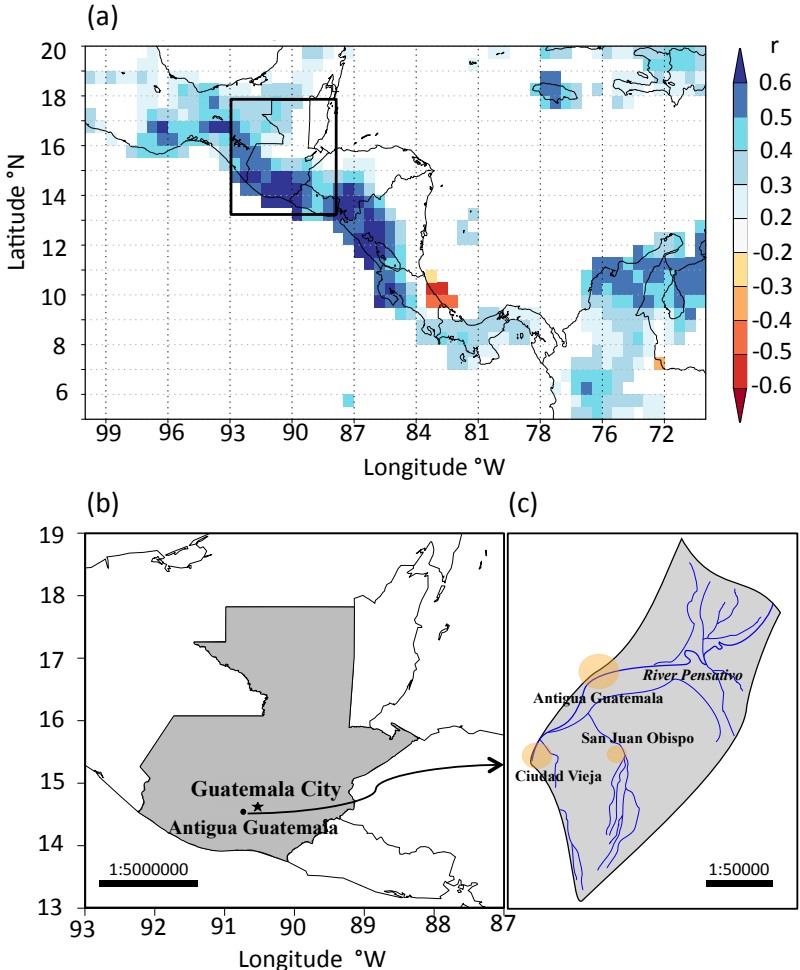

**Figure 1.** a. Spatial correlation between the Guatemala City Observatory precipitation record (INSIVUMEH, 1928-2012) and gridded precipitation across Central America (GPCC V7 reconstruction database; grid size 0.5°; Schneider et al., 2015) averaged for the May-October period and plotted according to strength of correlation coefficient. b. map of present-day
5   Republic of Guatemala and location of the cities of Antigua Guatemala (Santiago de Guatemala) and Guatemala City (Nueva Guatemala de la Asunción). c. map of the River Pensativo catchment relative to the city of Antigua Guatemala and towns of Ciudad Vieja and San Juan Obispo.





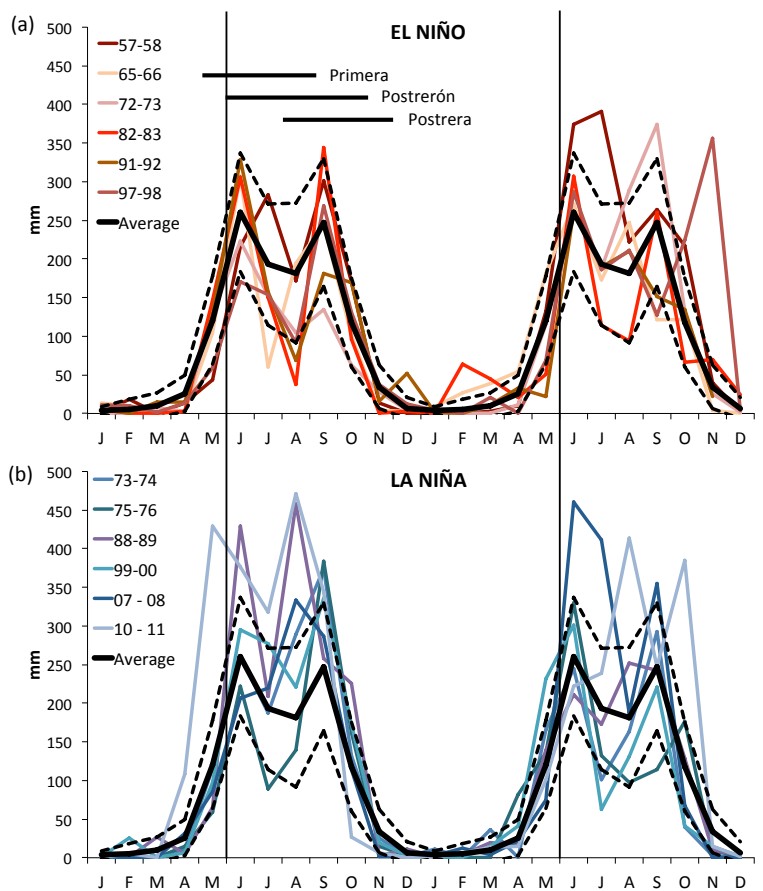

**Figure 2.** Average monthly precipitation for Guatemala City Observatory (INSIVUMEH) between 1960-1990 (black) a. compared to individual precipitation records of the 6 strongest El Niños (57-58, 65-66, 72-73, 82-83, 97-98) and b.

5    compared to the 6 strongest La Niñas (73-74, 75-76, 88-89, 99-00, 07-08, 10-11) of the 20[th] century in the Oceanic Niño Index (ONI; NOAA Climate Prediction Center (CPC)). Vertical lines mark the usual start and end of an El Niño Southern Oscillation (ENSO; May-June) year, peaking in boreal autumn and winter (Curtis, 2002). Horizontal lines indicate the three time frames of modern crop production in Central America (Arias et al., 2012). Dashed lines indicate one standard deviation with respect to the average monthly precipitation of the 1960-1990 period.





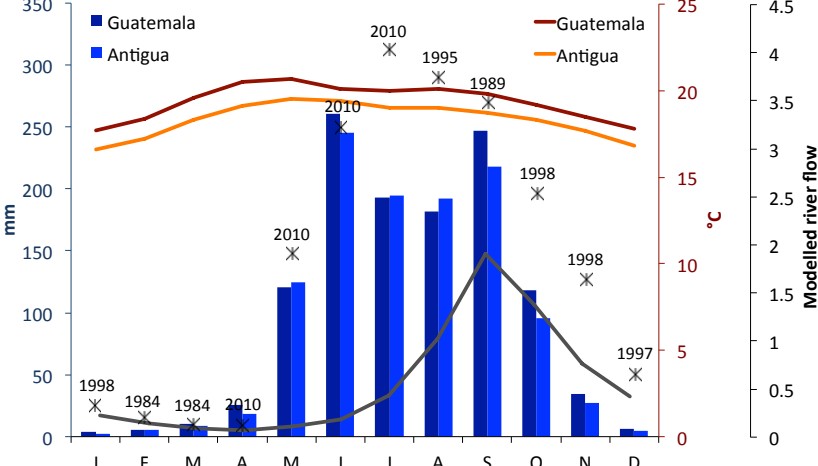

**Figure 3.** Averaged monthly rainfall and temperature for the cities of Guatemala and the closest meteorological station to Antigua Guatemala (called Suiza Contenta) for the period 1980-2010 (INSIVUMEH). Grey line shows an estimation of monthly average runoff per month of the Pensativo river for the 1981-2010 period based on results from the Orchidee land surface model. The asterisks show the highest estimated river flow per month and the year in which they happened. All the highest estimated river flow values of the rainy season correspond to La Niña years.



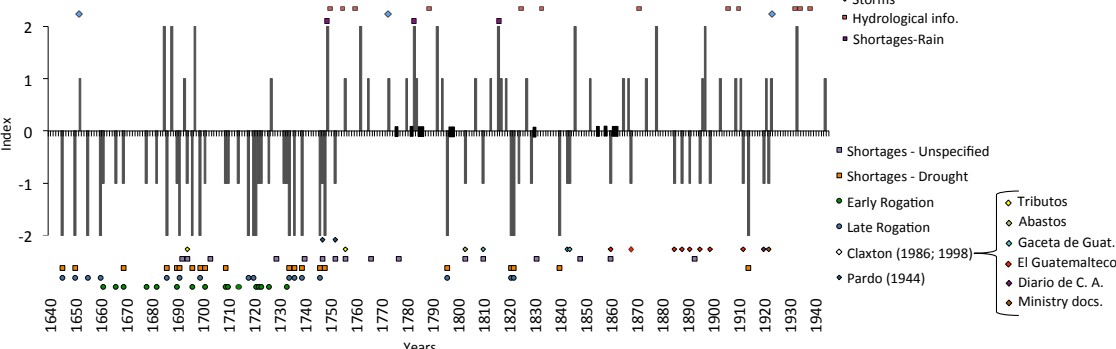

**Figure 4.** Five-category semi-quantitative hydrological index for the cities of Antigua Guatemala and Guatemala City starting in 1640 and ending in 1945. Rainfall in each year is classified as very dry = -2, dry = -1, normal= 0, wet = +1 and very wet = +2 based on the information obtained from the city and municipal council records of both cities of the Republic of Guatemala and complemented with the information from Claxton (1986; 1998) and Pardo (1944). All storms or *temporales*, early and late rogation ceremonies, years with hydrological information not considered sufficiently detailed to enable classification of the year as wet, and food shortages, whether or not the cause was specified, are indicated. Years in which the studies of Pardo (1944) and Claxton (1986; 1998) are used to reconstruct rainfall variability are also indicated. The black blocks mark where there are missing or damaged books and so no information is available.




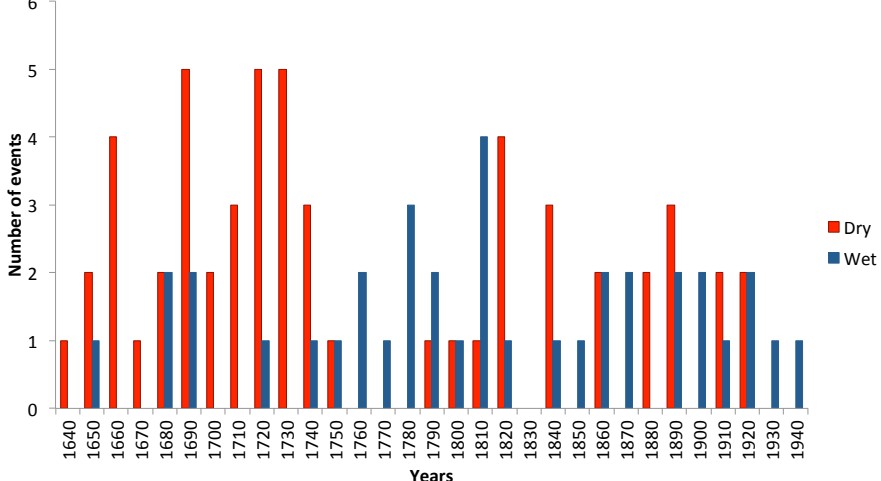

**Figure 5.** Number of dry and wet events per decade, based on information from city and municipal council records, Claxton (1986; 1998) and Pardo (1944).

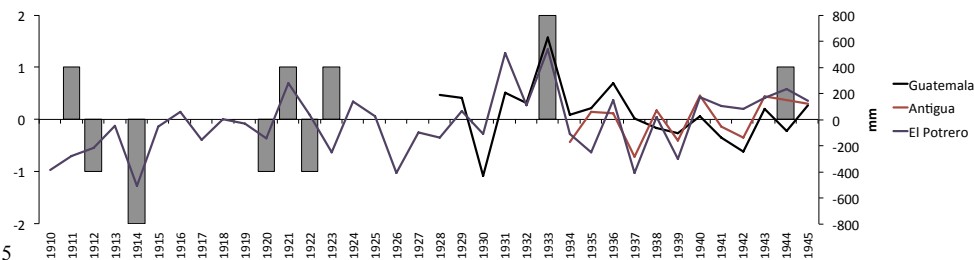

**Figure 6.** Semi-quantitative hydrological index reconstructed in this study (1910-1945) and precipitation anomalies (with respect to the 1934-1945 period) for the meteorological stations of Antigua Guatemala, Guatemala City and El Potrero.

