# Peer review of "300-years of hydrological records and societal responses to droughts and floods on the Pacific coast of Central America"

_Climate of the Past, 2017_

## Referee Comment (RC1) · D. Nash (Referee) · 27 Mar 2017

**General comments**

Many thanks for the opportunity to review this manuscript for Climate of the Past Discussions. The paper concerns the reconstruction of hydrological variability over a 300 year period for Guatemala, Central America, using documentary evidence in the form of city and municipal council records beginning in 1640. Given its focus on climate variability in the last 400 years, it is likely to be of considerable interest to the readership of CPD. The authors also make a compelling climatic argument for the results to be more widely representative of conditions along the western coast of Central America during the study period. The scientific approach adopted, and the methods applied, are entirely appropriate for the nature of the study and indicate rigour. The archival sources used for the study appear to be very rich in detail and have been explored exhaustively. The paper is extremely well written - I would go so far as to say that it is a pleasure to read - and the authors should be commended for this. It is well structured and very well organised (although note my comments below), and suitably illustrated with appropriate tables and figures. The results are discussed in good detail, and make reference to a wide but appropriate range of the previous literature. Overall, I have no reservations in seeing the paper published following the revisions recommended below.

Major point

My main concern with the manuscript is in relation to the results as they are introduced in section 4.1. It is fairly standard within historical climatology publications to provide the reader with an indication of the confidence with which each year in a reconstruction is classified. This could, for example, be in the form of a graphic showing the number of individual quotes used for each year of the reconstruction. If this approach were used, some recognition would, of course, be needed in the text to reflect that a single detailed quote describing climatic variability over a region might provide much more relevant climatic information than 10 less detailed quotes. If this type of graphic is not appropriate for this study, then, following Kelso and Vogel (2007), each year could instead be given a 'confidence rating' (from 1-3) to indicate the confidence with which the authors regard the classification for that year. Given the quality and quantity of material within the documentary sources, I would imagine that most years would be given a high confidence rating. However, lower confidence ratings could be used to highlight some of the concerns for individual years that the authors highlight within later sections of the manuscript (e.g. in the final sentences of section 4.2).

Kelso C, Vogel C (2007) The climate of Namaqualand in the nineteenth century. Climatic Change 83:357-380.

Minor points

1. The opening paragraph of section 3.1 discusses factors which could lead to crop loss or shortages. Given my experience of reading equivalent historical source materials for southern Africa, I am slightly surprised that there is no mention of either pestilence or conflict as having potential impact on harvest yields. If these factors are irrelevant for Central America, it might be useful to include some explanation as to why this is the case. This may need no more than an additional sentence.

2. To assist the reader, the opening paragraph of section 3.4 could make greater cross reference into table 1.

3. When I first read section 4.1 of the script, I was slightly surprised that no qualitative descriptions of, for example, the wettest and driest years within the reconstruction period were included. A few illustrative quotes, for example, would add real "colour" for the reader to an otherwise fairly brief and "dry" section. I then read on and discovered that descriptions of individual events were embedded within section 5.1 under the heading of "Discussion". I appreciate that some of the quotes are used to highlight discrepancies or difficulties in classifying individual years (see my major point above). However, the authors might consider a little reorganisation to move some of the descriptive material from section 5.1 into section 4.1.

4. The second paragraph of section 5.3 notes that the recent observations that warm ENSO conditions are associated with lower precipitation might be a non-stationary relationship. This is not the first time that this sort of discrepancy has been identified within historical climatology studies. The authors should refer to Adamson and Nash (2014) or Ashcroft et al. (2016) as entry points into this literature.

Adamson GCD, Nash DJ (2014) Documentary reconstruction of monsoon rainfall variability over western India, 1781-1860. Climate Dynamics 42:749-769.

Ashcroft, L., Gergis, J. and Karoly, D.J., 2016. Long-term stationarity of El Niño–Southern Oscillation teleconnections in southeastern Australia. Climate Dynamics, 46(9): 2991–3006.

---

## Short Comment (SC1) · 5 May 2017

he PAGES Data Stewardship Integrative Activity seeks to advance best practices for sharing data generated and assembled as part of all PAGES-related activities. As part of this activity, a team of reviewers has been constituted for the "Climate of the Past 2000 years" Special Issue. The data team is reviewing the data handling within each of the CP-Discussion papers in relation to the CP data policy and current best practices. The team has identified essential and recommended additions for each paper, with the goal of achieving a high and consistent level of data stewardship across the 2k Special Issue. We recognize that an additional effort will likely be required to meet the high level of data stewardship envisaged, and we appreciate dedication and contribution of the

authors. This includes the use of Data Citations (see example in supplement). We ask authors to respond to our comments as part of the regular open interactive discussion. If you have any questions about PAGES Data Stewardship principles, please contact any of us directly.

Best wishes for the success of your paper,

2k Special Issue Data Review Team (Darrell Kaufman, Nerilie Abram, Belen Martrat, Raphael Neukom, Scott St. George) and ex-officio team members (Marie-France Loutre, Lucien von Gunten)

Essential additions for this paper:

(1) Add a "data availability" section that describes where the primary input data (documents and weather station measurements) can be accessed (name and location of the archives), plus the output data from this study (#2 below).

(2) Submit the primary outcome of the data analyses to a public repository and include the Data Citation. This includes: (a) The hydrological index (y-axis value), type (plot symbol), location, year/season for each documented event in Fig 4. (b) The number and the source of information for each event in Fig. 5.

Recommended:

(3) We strongly encourage the authors to archive the meteorological datasets that were used to compare the reconstruction, especially the precipitation anomalies shown in Fig. 6.

Please also note the supplement to this comment:
http://www.clim-past-discuss.net/cp-2017-30/cp-2017-30-SC1-supplement.pdf

---

## Referee Comment (RC2) · Anonymous Referee #2 · 10 May 2017

General comments

This paper examines hydrological records, societal responses, and the relationships between the two in today's Antigua Guatemala and Guatemala City, over the period from 1640 to 1945. The basis for the presented semi-quantitative hydrological indices is derived from documentary data in the records of the city and municipal council meetings. These types of sources include descriptions of exceptional meteorological events such as heavy rain, flooding, dry weather conditions, and drought; as well as records of rogation ceremonies, crop shortages, etc. The indices derived for this paper cover the annual rainy season (May to October) and use a five-degree scale (very wet to very dry). There are exceptional periods that the authors present in their results, such as

the dry period from 1640 to 1740, as well as those in the 1820s and 1840s. The period between 1760 and 1810 was wet. The authors bring these periods together with the Little Ice Age and the variability of the Intertropical Convergence Zone (ITCZ) and the El Niño Southern Oscillation (ENSO).

I agree with David Nash that this is a very nice and convincing paper. The topic is clearly in the focus of Climate of the Past. The authors very clearly describe the historical context of the outstanding documentary sources which they used for the paper. They also make clear which methods they applied in order to reconstruct precipitation. As a result, the authors have presented a remarkable continuous hydrological index encompassing more than 300 years, with almost no gaps. Nonetheless, I wish to make a few remarks:

Specific comments

In the abstract, the authors refer to the onset of the Little Ice Age as coinciding with the dry period from 1640 to 1740. Later in the paper, on p. 16, they define the Little Ice Age as the period between 1500 and 1800. Both definitions are in use, but I would recommend rephrasing one of the two sentences so that there is no inconsistency in that regard.

Also in the abstract, the authors mention the complexity of the relationship between hydrological extremes and societal responses. In the paper, they do not discuss this complexity any further; perhaps a short explanation could be added.

In the paper, the description of the state of the art is divided between several chapters. Perhaps a short paragraph in the introduction, instead of the short presentations of literature in each chapter, would be useful. So, for instance, on p. 2 the authors mention previous research reconstructing rainfall and extreme events based on ecclesiastical records in Latin America and Spain. At this point, Rodrigo and Barriendos, who published important results on this subject, are not mentioned. They only appear on page 6 when the topic is discussed a second time.

[Figure]

The paper clearly describes why the records of the city and municipal council meetings are reliable and very useful for the reconstruction of hydrological variation. On p. 7 there are, in addition, three further reconstructions (Claxton 1986, 1998; Pardo 1944) mentioned as being based on "discontinuous primary sources and secondary sources". I recommend adding a short explanation of why these types of reconstructions and data collections are not so reliable as the results presented here.

The authors present very nice source examples on p. 7–9, but they do not give many comments on these examples. Perhaps fewer examples with more in-depth comments would be sufficient and more focussed.

The authors could also consider whether they want to gather all source-relevant information in one place. Currently, this information is distributed over several places. For instance, the reader gets the information that the Actas de Cabildo and the Actas Municipales are preserved in 190 volumes in the abstract, but in the main text this information only appears on p. 9, long after all the other source-relevant information. This information shows how much work was invested into this reconstruction. Therefore, it could be presented a bit earlier and more prominently in the introduction.

I congratulate the authors on these striking results and on a very informative and convincing paper, which I recommend publishing with a few minor revisions.

---

## Author Comment (AC1) · 15 Jun 2017

Dear David Nash,

We appreciate your generous comments on our manuscript.

Regarding your specific comments:

Major point: You are right to point out that we have not provided a clear indication of our confidence in the record for each year of our reconstruction. We will address this issue in the final version of the manuscript, and include a confidence rating for each year based on Kelso and Vogel (2007).

Minor points:

1. The early period of Spanish-indigenous contact will certainly have had an impact on harvest yields due to conflict, epidemics, and demographic decline (e.g., MacLeod, 2010). In the earliest records we used date to the 1640s; we identified outbreaks of epidemics (flu, typhus, measles, smallpox and cholera) which we expected to be associated with anomalous climatic conditions, but we found no significant relationship between these, shortages and the hydrological index reconstructed in our study. We aim to return to this question in future research. The Actas also record locust plagues, reducing yields in certain years, however the source regions of these plagues are distal and varied and the environmental triggers complex, so we did not include these as an indication of the local climatic situation. Further, these causes of the harvest loss were clearly attributed in the documents.

2. We agree.

3. We agree.

4. We agree – we should have better referenced this statement since it is a well recognised feature of ENSO teleconnections.

MacLeod, M. J.: Spanish Central America: A Socioeconomic History, 1520–1720, University of Texas Press, 2010.

Kelso C., Vogel C.: The climate of Namaqualand in the nineteenth century, Climatic Change, 83:357-380, 2007.

---

## Author Comment (AC2) · 15 Jun 2017

Dear 2k Special Issue Data Review Team,

Thank you for advancing the best practice for sharing and archiving of published data. This is a very important consideration that we whole-heartedly support.

We will make our data available online (using the NOAA National Centers for Environmental Information https://www.ncdc.noaa.gov/data-access/paleoclimatology-data/datasets) and will include all the datasets requested.

---

## Author Comment (AC3) · 15 Jun 2017

Thank you very much for taking the time to review our manuscript and for the positive comments regarding our hydrological reconstruction. We agree with all your specific comments and minor suggested changes. We appreciate your help with improving the paper and would welcome acknowledging you personally in our final manuscript.

---

## Author Response (AR1)

**Alvaro Guevara-Murua**
**School of Earth Sciences**
Wills Memorial Building
Queens Road, Bristol, BS8 1RJ
United Kingdom
E-mail: alvaro.guevara.2012@my.bristol.ac.uk

*Climate of the Past*
Dr Raphael Neukom
Institute of Geography
University of Bern
11 October 2017

**Re: " 300-years of hydrological records and societal responses to droughts and floods on the Pacific coast of Central America" (cp-2017-30)**

Dear Dr Neukom,

On behalf of myself and my co-authors, I would like to thank you and the Reviewers again for the positive comments and helpful suggestions for improving our manuscript. Below we outline how we have addressed their comments and incorporated their suggestions.

Please let us know if you have any questions.

Sincerely,

**Reviewer #1, David Nash:**

**A) Major point**

1. Results as introduced in section 4.1/indication of the confidence with which each year in a reconstruction is classified: a 'confidence rating' (from 1-3) could be given following Kelso and Vogel (2007).

> 1.  As recommended, we have included a confidence rating for each year based on Kelso and Vogel (2007). Further information can be found in section 3.5 and in the supplementary information.

**B) Minor points**

1. Factors which could lead to crop loss or shortages/potential impact of pestilence or conflict on harvest yields

2. The opening paragraph of section 3.4 could make greater cross reference to table 1.

3. Section 4.1 of the script: the authors might consider a little reorganisation to move some of the descriptive material from section 5.1 into section 4.1.

4. Second paragraph of section 5.3: the authors should refer to Adamson and Nash (2014) or Ashcroft et al. (2016) as entry points into this literature.

1.  We identified in our sources reports of outbreaks of epidemics (flu, typhus, measles, smallpox and cholera), which we expected would be associated with anomalous climatic conditions. However, as we found no significant relationship between these and anomalous climatic conditions we have inserted a new paragraph discussing this issue (starting on line 24 of page 9).
2.  The opening paragraph of section 3.4 now cross-references Table 1
3.  We have moved some of our descriptive materials from section 5.1 to section 4.1.
4.  The second paragraph of section 5.3 is now more fully referenced.

**Reviewer #2, Darrel Kaufman:**

**Essential additions:**

1. A "data availability" section that describes where the primary data (documents and weather station measurements) can be accessed.

2. Primary outcome of the data analyses should be submitted to a public repository and should include the Data Citation. This includes: (a) The hydrological index (y-axis value), type (plot symbol), location, year/season for each documented event in Fig 4. (b) The number and the source of information for each event in Fig. 5.

**Recommended:**

1. Archiving of the meteorological datasets that were used to compare the reconstruction, especially the precipitation anomalies shown in Fig. 6.

1. A data availability section has been included at the end of the manuscript, indicating the locations where documentary sources and weather station measurements can be accessed.

2. In addition, our data will be available online shortly (using the NOAA National Centers for Environmental Information https://www.ncdc.noaa.gov/data-access/paleoclimatology-data/datasets). All datasets requested have been prepared following the NOAA templates and have been sent to the correspondent email. As soon as we receive the link to the datasets, it will be included in the manuscript.

3. The meteorological datasets used in this study have been already archived at the KNMI Climate Explorer, and can be accessed at the following webpage: https://climexp.knmi.nl/selectstation.cgi?id=someone@somewhere

**Reviewer #3:**

**Specific comments:**

1. The Little Ice Age definitions: rephrasing of sentences for consistency.

2. The complexity of the relationship between hydrological extremes and societal responses is not fully followed-up in the paper.

3. Previous research reconstructing rainfall and extreme events based on ecclesiastical records in Latin America and Spain, specifically Rodrigo and Barriendos, should be referenced in the Introduction.

4. A short explanation of why reconstructions and data collections based on discontinuous sources are not as reliable as the results presented her should be considered.

5. Source-relevant information should be made more prominent in the text – e.g. that the Actas de Cabildo and the Actas Municipales are preserved in 190 volumes is mentioned in the abstract, but in the main text it only appears on p. 9. This information shows how much work was invested into this reconstruction.

1. The end of the Abstract has been modified to be consistent with the Little Ice Age dates.
2. We have clarified the types of impacts that our sources enable us to reconstruct and the additional materials that could be consulted for a more extended discussion of impacts and responses.
3. This has been addressed in lines 22 and 23 of page 3.
4. We now explain in more detail why discontinuous sources are less reliable (see paragraph starting in line 5 of page 10).
5. We have made reference more explicitly and earlier in the paper to the sources we have used for the study.

---

## Author Response (AR2)

**Alvaro Guevara-Murua**

**School of Earth Sciences**

Wills Memorial Building

Queens Road, Bristol, BS8 1RJ

United Kingdom

E-mail: alvaro.guevara.2012@my.bristol.ac.uk

*Climate of the Past*
Dr Raphael Neukom
Institute of Geography
University of Bern
30 October 2017

**Re: " 300-years of hydrological records and societal responses to droughts and floods on the Pacific coast of Central America" (cp-2017-30)**

Dear Dr Neukom,

On behalf of myself and my co-authors, I would like to thank you and the Reviewers again for the positive comments and for accepting our manuscript for publication. Below we outline how we have addressed your comments and incorporated your suggestions.

Please let us know if you have any questions.

Sincerely,

**(A): Data stewardship:**

**A1: There seems to be a contradiction between the reply to the editor and the information in the data availability section regarding the met' station data. Rather than referring to the web-search interface (which is not permanent), the authors should cite the data behind the search.**

> The data are now stored as GHCN stations, so we have provided station names and number that allow identification of the stations within the GHCN v2 database.

**A2: NOAA WDC Paleo link: Please provide the link to the editor once it is available so, that the data availability can be reviewed by the data stewardship team of the special issue editorial board. The paper will be accepted once we've had a look at the data.**

> We have provided the link from the NOAA in order for the data to be reviewed.

**(B) Reviewer #2:**

**B1: One technical correction is needed (p.23, line 12) - I suspect that authors mean 'stationary' rather than 'non-stationary' here.**

> We actually meant non-stationary in p.23, line 12 of previous manuscript version. However, we appreciate that this paragraph was not clear enough, so we have rewritten it appropriately in this version.

**(C) Editor comments:**

**C1: Please remove "[through the ###2k working group]" From the acknowledgments. I suggest to just write: "This is a contribution to the PAGES 2k Network".**

**C2: Consider acknowledging the reviewers.**

Apologies. We have included the reviewers in the acknowledgements section and re-phrased the reference to the PAGES 2k Network.

[revised manuscript text omitted]